# Study of Variable Thickness Magnetorheological Transmission Performance of Electrothermal Shape Memory Alloy Squeeze

**Song Chen, Wenjian Chen** 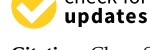 **and Jin Huang \***

College of Mechanical Engineering, Chongqing University of Technology, Chongqing 400054, China; songchen@cqut.edu.cn (S.C.); chenwenjian@2020.cqut.edu.cn (W.C.)
\* Correspondence: jhuangcq@cqut.edu.cn

**Abstract:** This paper designs a new composite transmission device for improving the transmission torque by squeezing magnetorheological fluid (MRF) with an electrothermal shape memory alloy (SMA) spring. Based on the finite element method, a numerical analysis of the magnetic circuit and magnetic field distribution of the magnetorheological (MR) transmission is presented, as well as a theoretical derivation and calculation of the squeezing force output by the electrothermal SMA spring and the transfer torque of the variable thickness MR transmission. In addition, the output characteristics of the electrothermal SMA spring at different temperatures are analyzed, as are the torque characteristics of the variable thickness MR transmission. The research shows that the electrothermal SMA springs exhibit a highly non-linear squeezing force output during the temperature rise, and the increase in current affects the martensite phase transition quality, and thus the phase transition temperature. The squeezing force generated by the springs increases significantly when the temperature is within the martensitic phase transformation interval, with a maximum squeezing force of 318.43 N when the SMA temperature reaches 100 °C. The proposed variable thickness MR transmission can increase the maximum torque by 4.88 times under SMA spring squeezing force, and its maximum transmitted torque is increased from 15.08 to 73.56 N·m. By squeezing the MRF with an electrothermal SMA spring, the torque of the variable thickness MR transmission can be increased quickly and effectively.

**Keywords:** electrothermal shape memory alloys; magnetorheological fluid; squeeze; variable thickness; torque

## 1. Introduction

Magnetorheological fluid (MRF), as a kind of smart material with solid–liquid two-phase whose rheological properties are controlled by an external magnetic field, is a non-Newtonian fluid composed of additives, base fluid, and micron-sized magnetic particles uniformly distributed in the base fluid [1]. Under the effect of the external magnetic field, MRF can complete the reversible transformation from a liquid to a viscoplastic body in milliseconds, and the shear yield stress and apparent viscosity of the material can vary steplessly with the external magnetic field. With the characteristics such as rapid response, reversible transformation, and controllable characteristics, MRF has a very wide range of applications in the transmission field [2,3]. Shape memory alloy (SMA) is a type of smart alloy material with a shape memory effect and super-elasticity [4]. In case of temperature changes, the internal metallographic structure of SMA switches between non-twin martensite and high-temperature austenite, and the external morphology of the alloy changes accordingly depending on the internal crystal structure. Due to the unique shape memory effect, SMA is widely applied in fields such as medical, control, machinery, and energy [5,6].

Researchers have performed extensive studies on the material properties of MRF and SMA, as well as their applications in the field of smart transmission. Huang et al. [7] presented the operational principle of the cylindrical MR fluid brake, and derived an

engineering expression for the torque to provide the theoretical foundations in the design of the cylindrical MR fluid brake. Based on this equation, the volume and thickness of the annular MR fluids within the brake are expressed as functions of the desired ratio of torques with a saturated magnetic field and without an external field, the controlled mechanical power, and the MR fluid material properties. Qin et al. [8] designed a multi-cylinder MR transmission device, and the torque/volume ratio of the device was 27.864 kN/m$^2$ based on experiments, with the MRF transmitting torque in multiple cylindrical gaps. The torque density of the MR device was greatly improved. Xiong et al. [9] investigated the temperature distribution of MR devices and the effect of the coil and air gap dimensions on MRF performance. The relationship between coil current and the MRF temperature was quantified in this paper by modeling the heat generation of the coil. Dai et al. [10] present a novel composite magnetorheological fluid clutch design that is based on both disc- and cylinder-type clutches. The transmittable torque can be 1.45 times larger than the existing disc-type clutches with the same input plate diameter. Wang et al. [11] design, simulate, and experimentally investigate a magnetorheological (MR) brake under compression-shear mode and establish mathematical torque expressions, operating under compression-shear mode assuming the Herschel–Bulkley model. The results show that the large torque could be produced at high applied currents, high compressive stress, large compressive strain, and small initial gap distances. Hegger et al. [12] deal with the squeeze strengthening effect, which causes an enhancement in the shear stress of magnetorheological fluids. A concept for a new magnetorheological fluid test-actuator enabling a self-induced squeeze strengthening effect through an eccentric shear gap design by two independent rotations is presented and theoretically investigated. Stachowiak et al. [13] present the design strategy for a system composed of SMA spring and steel spring, and calculate and measure the force versus stroke characteristics of the designed system at high- and low-temperature conditions. The electro-thermo-mechanical characterization of SMA spring has been carried out. Hwang et al. [14] introduced a design concept for an SMA rotary actuator that transforms the linear motion of SMA into a high-torque rotary motion and explains its thermodynamic behavior based on experiments. The actuator is realized with a rotational driving mechanism devised on the basis of the operating principle of wobble-stepping motors. In designing the actuator, the analysis result of the parametric effect is utilized, and driving characterization and working performance verification are experimentally carried out with a fabricated functional prototype. With the experimental results, differentiating characteristics related to the operating principle of the proposed actuator and the thermomechanical behavior of SMA elements are investigated and discussed. Andronov et al. [15] designed an SMA spring design method for a given range of stiffness. Based on the physical and mechanical properties of titanium nickelide, the relationship between spring shear strain rate and output stress was quantitatively investigated.

However, the above studies mainly focus on the characteristics and application areas of a single MRF or SMA, while they are less involved in the combined use of MRF and SMA. To address the problem of magnetorheological fluid performance degradation at high temperatures, Chen et al. [16] proposed an SMA and multi-arc surface MR composite transmission device. The SMA spring adds torque by pushing the friction rings, allowing the MR transmission to maintain a smooth transmission over the entire operating temperature range.

The magnetic saturation of MRF limits its transmission torque so that it cannot be continuously increased, and the transmission torque based on shear mode design is relatively small, thereby limiting the application and development of MRF in the transmission field. Furthermore, the SMA devices mentioned above generally lack active control of the output force. In response to such problems, this article proposes an electrothermal SMA spring-driven variable thickness MRF composite transmission device. The heat generated by the copper-core polyurethane enameled wire spirally wound on the SMA spring causes the spring to be heated and elongated, which drives the squeeze disc to push the MRF axially, changing the thickness of the working gap and finally achieving the purpose of

squeezing the MRF. The shear yield stress of the MRF is significantly increased by the squeeze strengthening effect, thus significantly increasing the torque of the transmission.

## 2. Materials and Methods

### 2.1. Working Principles of Variable Thickness MRF Transmission Device

The working principles of the variable thickness MRF transmission device actuated by electrothermal SMA are shown in Figure 1. This MRF transmission device mainly consists of components such as an input shaft, output shaft, squeeze disc, rotary housing, electrothermal SMA spring, guide disc, and excitation coil. The electrothermal SMA spring is positioned on the guide plate, and the two ends of the spring are respectively connected with the rotary housing and the guide plate in a fixed manner. The output shaft is bolted to the rotary housing, the guide disc and the squeeze disc are connected by the bolts, and the brush slip ring is at the shoulder of the output shaft. The wires of the brush slip ring are connected to the excitation coil and the electrothermal SMA spring, respectively, through the wire holes on the rotary housing, and the power is supplied to the coil and the electrothermal SMA spring, respectively, through the external slip ring. The MRF is filled between the rotary part of the input shaft and the squeeze disc, which is sealed by sealing rings on each side. For the purpose of changing the thickness of the working gap of the MRF transmission device during operation, a squeeze disc is installed between the disc area of the input shaft and the rotary housing, and the squeeze disc is propelled by the displacement or force output by the electrothermal SMA spring during the temperature rise, thereby enabling changes in the distance of the MRF working gap formed by the squeeze disc and the disc area of the input shaft, so that the process of changing the thickness of the MRF working gap can be controlled by an applied current.

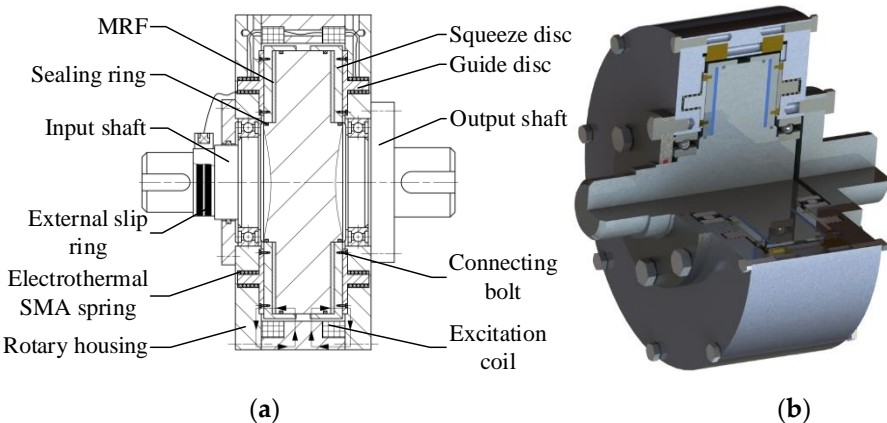

(**a**) (**b**)

**Figure 1.** Working principles of MRF transmission device. (**a**) 2D structure diagram and (**b**) 3D structure diagram.

The operating principles of the electrothermal SMA spring-driven variable thickness MRF transmission device are as follows:

(1) Under the initial condition, when the coil is not energized, the MRF is in zero-field state, and the input shaft cannot drive the output shaft to rotate only by the shear stress generated by the MRF zero-field viscosity during the rotation.

(2) When the transmission device starts to function, the excitation coil is energized to generate a magnetic field, the magnetic particles in the MRF are arranged in a chain along the magnetic flux direction, the shear yield stress is significantly enhanced, and the output shaft is rotated by this shear yield stress; furthermore, as the coil current increases, the transmission torque of the device can be further increased.

(3) To further increase the torque of the MRF transmission device, by loading the electrothermal SMA spring with current, the Joule heat generated by the current generates a shape memory effect in the SMA, and the force or displacement generated by the

electrothermal SMA drives the squeeze disc to gradually start squeezing the MRF working gap thickness from 1.5 mm, and the axial squeeze increases the MRF shear yield stress exponentially. As a result, the transmission torque of the MRF transmission device is further increased.

(4) At the end of the operation, the excitation coil is de-energized and the MRF is restored to a Newtonian fluid upon the disappearance of the magnetic field by the viscoplastic body, and the electrothermal SMA spring gradually regains its original length as the temperature drops, and the output shaft of the drive stops rotating.

### 2.2. Structural Dimensional Design of the Transmission

To study the magnetic field strength upon the variation of MRF working gap thickness and to ensure that the MRF chaining direction is perpendicular to the MRF shear plane, the magnetic circuit is replaced equivalently based on the equivalent circuit method [17]. By simplifying the MRF transmission device to a two-dimensional axisymmetric model, a simplified magnetic circuit model consisting of a single-side cross-section is shown in Figure 2.

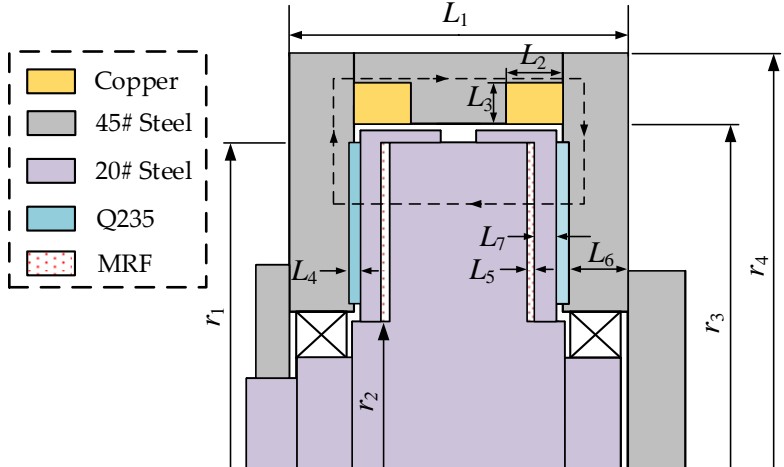

**Figure 2.** Magnetic circuit model of variable thickness MRF transmission device.

We can obtain the structural dimensions from Table 1. The magnetomotive force, $F_{\mathrm{m}}$, was used as the current excitation in the finite element analysis (FEA):

$$F_{\mathrm{m}} = NI \tag{1}$$

where $N$ refers to the number of turns of the coil and $I$ refers to the current loaded in the coil.

**Table 1.** Structural parameters of the transmission device.

| $r_1$ | $r_2$ | $r_3$ | $r_4$ | $L_1$ | $L_2$ | $L_3$ | $L_4$ | $L_5$ | $L_6$ | $L_7$ |
|-------|-------|-------|-------|-------|-------|-------|-------|-------|-------|-------|
| 55 mm | 25 mm | 58 mm | 70 mm | 57 mm | 9.5 mm | 6 mm | 2 mm | 1.5 mm | 10 mm | 3.5 mm |

To improve the calculation accuracy of the magnetic field characteristics, consideration is given to the nonlinear magnetic permeability of the material in the finite element analysis. The magnetization curves of the materials used in the transmission are shown in Figure 3, which were imported into the FEA software.

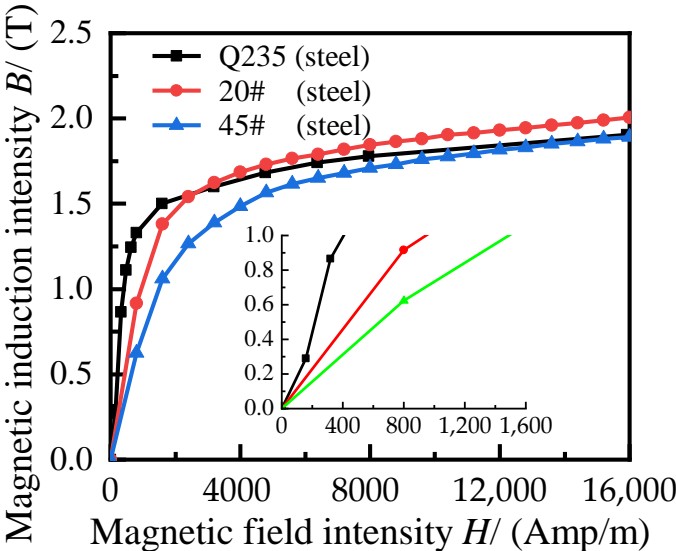

**Figure 3.** Magnetization curves of materials.

*2.3. Squeeze Strengthening Effect of the MRF*

The material properties, as well as the magnetization curve and magnetic flux density–shear yield stress curve, for MRF-J01T provided by MRF for Chongqing Instrument Materials Research Institute, are shown in Figure 4.

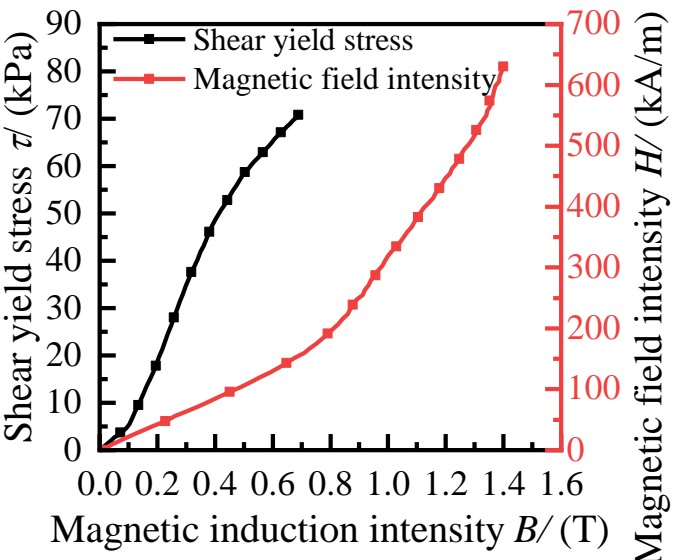

**Figure 4.** MRF (MRF-J01T) material characteristic curve.

Based on the performance parameters of MRF in Figure 2, the relations between the magnetic induction strength, *B*, and the shear yield stress before squeeze, $\tau_y(B)$, can be obtained with the fitting equation, as follows:

$$\tau_y(B) = 965.5B^4 - 1651B^3 + 86.17B^2 - 23.99B + 0.9684 \tag{2}$$

According to the experiments of Liu et al. [18], it is known that the shear yield stress of the magnetorheological fluid increases exponentially when it is subjected to axial squeezing pressure. Among them, when the magnetorheological fluid is shear-deformed, the internal deformation of the magnetic chain is as shown in Figure 5.

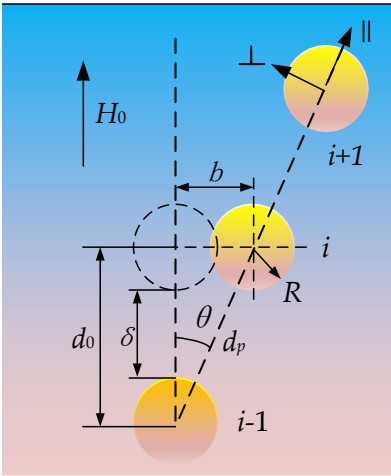

**Figure 5.** Particles in a magnetic chain.

The shear yield stress of the MRF can be mainly divided into two parts: the magnetically induced yield stress of the modified dipole model, $\tau_m$, and the friction yield stress by tribology, $\tau_f$. The additional shear stress approximation caused by the applied magnetic field can be obtained from [19]:

$$\tau_m = 3\phi\mu_f\mu_0\beta^2H_0^2\left(\frac{R}{d_0}\right)^3\zeta\left(\left(\frac{10}{A^2}+\frac{2}{B^2}\right)+\frac{48\beta s}{A^3}\left(\frac{R}{d_0}\right)^3\right)\gamma \quad (3)$$

where $A = 1 - 4\beta\cos^3\theta(R/d_0)^3\zeta$, $B = 1 + 2\beta\cos^3\theta(R/d_0)^3\zeta$, $\zeta = 1.202$, $\beta = (\mu_p - \mu_f)/(\mu_f + 2\mu_p)$, $H_0$ is the initial magnetic field strength, $R$ is the radius of magnetic particles, and $d_0$ is the distance between two adjacent magnetic particles. The remaining parameters are shown in Table 2.

**Table 2.** MRF (MRF-J01T) material performance parameters.

| Parameter | Value | Parameter | Value |
|---|---|---|---|
| Volume fraction ($\phi$)/% | 25 | Zero-field viscosity ($\eta$)/Pa·s | 0.8 |
| Density/g/cm$^3$ | 2.65 | Base liquid | Silicone oil |
| Relative permeability of the medium ($\mu_f$) | 400 | Relative permeability of particles ($\mu_p$) | 10 |
| Vacuum permeability ($\mu_0$)/T·m/A | $4\pi \times 10^{-7}$ | | |

From Figure 5, the shear strain, $\gamma$, of the magnetic chain can be expressed as:

$$\gamma = \frac{b}{d_0} = \tan\theta \quad (4)$$

Assume that the length of the magnetic chain is constant in the case of shear deformation, then the relationship between the shear strain, $\gamma$, and the adjacent particles' distance, $\delta$, within the chain can be expressed according to Equation (4), as [20]:

$$\frac{\delta}{R} = 2(\sqrt{1+\gamma^2}-1) \quad (5)$$

From Equations (3)–(5), when $H_0 = 100$ kA/m, the shear yield stress is influenced by the shear strain, and the change trend is shown in Figure 6. The magnetic yield stress rises rapidly as the shear strain increases when the shear strain is less than 0.5. However, when MRF produces large deformation, the magnetic chain will break and the shear yield stress will be reduced, and the MRF regains its fluidity.

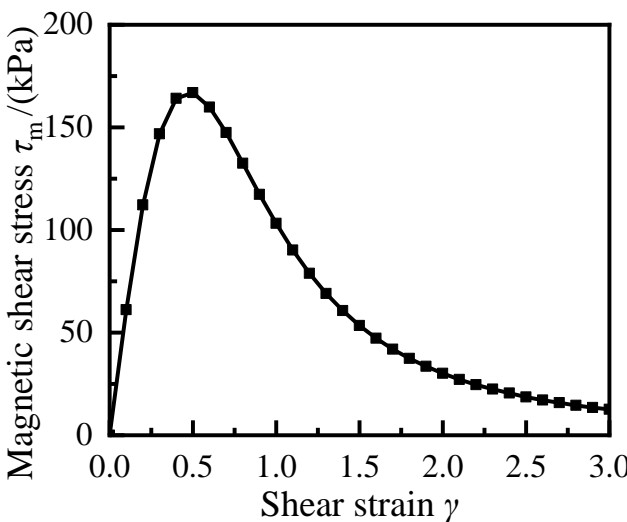

**Figure 6.** Effect of shear strain on dipoles' shear yield stress.

Now, the MRF is in the case of deformation under axial extrusion, the single chain becomes shorter, bends, and finally aggregates into a more compact and robust BCT structure, in which the chain behaves as low-yield strain and high-yield stress [21]. As the strain increases, the height of the chain decreases, as shown in Figure 7.

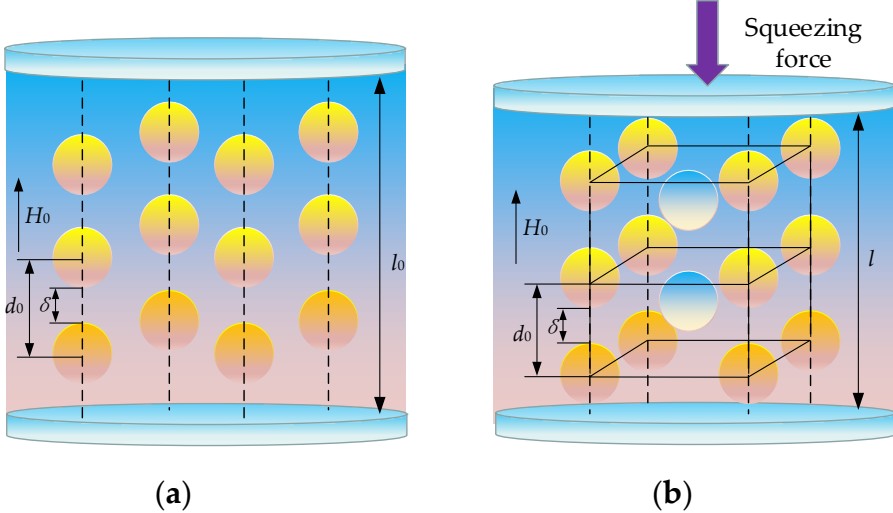

**Figure 7.** The magnetic chain model of the lattice structure. (**a**) Magnetic chain before squeeze, and (**b**) magnetic chain after squeeze.

In the process of lattice transformation, the relationship between the axial strain of the MRF and the particle concentration can be expressed as:

$$\varepsilon = \frac{V/\phi_{P1} - V/\phi_{P2}}{V/\phi_{P1}} = \frac{l_0 - l}{l_0}$$
$$\frac{R}{d_0} = X\sigma + Y$$

$$(6)$$

where $V$ is the total particle volume of MRF before and after squeezing, $\phi_{p1}$ and $\phi_{p2}$ are the particle volume concentrations, respectively, and X and Y are the squeeze deformation constants after yielding, which can be measured experimentally.

When the MRF are compressed, the influence of friction should be considered. In fact, when plastic deformation occurs because of heavy loading, the contact area increases

linearly with the load. This normal compression will increase the friction between the magnetic particles. Therefore, the tribology shear yield stress, $\tau_f$, can be expressed as:

$$\tau_f = \frac{C_M}{\left[\alpha(1 - C^2{}_M)\right]^{1/2}}\sigma \tag{7}$$

where $\sigma$ is the squeeze stress on MRF, and $C_M$ and $\alpha$ are two correction coefficients about the deformation of MRF-J01T; in this article, we assume $C_M = 0.3$ and $\alpha = 9$.

Combining the two above considerations, the yield stress after squeeze, $\tau_y{}'(B)$, can be expressed as:

$$\tau_y'(B) = K_1\tau_m + K_2\tau_f \tag{8}$$

where $K_1$ and $K_2$ are contribution coefficients of $\tau_m$ and $\tau_f$, respectively.

If particles are not in contact, $K_1 = 1$ and $K_2 = 0$. The internal friction increases slightly with the magnetic field. Here, its contribution coefficient is estimated empirically as:

$$K_2 = \frac{1}{4}\left[\mathrm{sgn}(B - B_0)\left(1 - e^{(-|B-B_0|/\Delta)}\right) + 3\right] \tag{9}$$

where $B_0$ is the squeeze strengthening effect's critical magnetic induction intensity point, and $\Delta$ is the intensity of the strength change.

### 2.4. Magnetic Field Finite Element Simulation

Based on the simplified model of the variable thickness MRF transmission device shown in Figure 2, a two-dimensional axisymmetric finite element model is established, and the calculated results of the magnetic circuit analysis are adopted as the magnetic field excitation conditions of the finite element model.

The area surrounding the MRF transmission device magnetic field analysis model is set to air with a relative permeability of 1. The number of turns of the coil is set to 100 turns, and the current directions of both coils are the same. At different working gap thicknesses, the magnetic field distribution in the variable thickness MRF transmission device derived from the finite element steady-state analysis is shown in Figure 8, where the magnetic lines of force all pass perpendicularly through the MRF working area, enabling the MRF to generate the maximum shear yield stress.

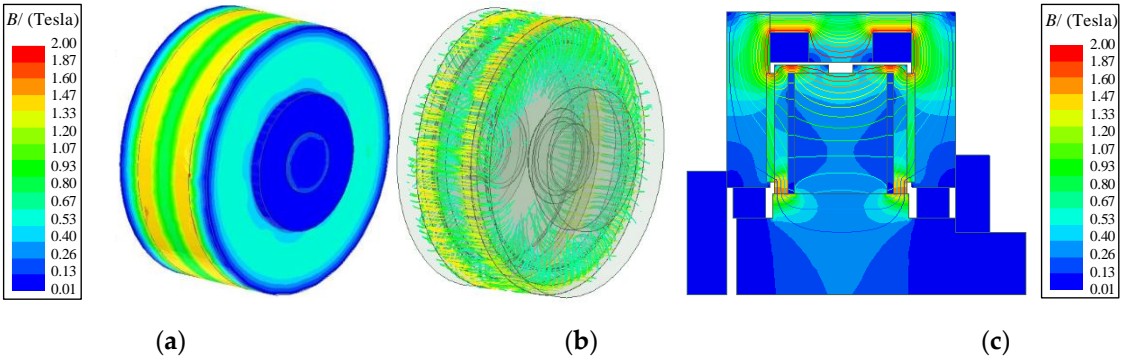

**Figure 8.** Cloud chart of magnetic induction intensity and magnetic lines of force at the current of 3.0A. (**a**) 3D cloud chart, (**b**) streamline of the magnetic induction intensity, and (**c**) magnetic field distribution on an arbitrary cross-section.

As shown in Figure 8, at a current $I = 3.0$ A and a gap thickness, $h$, of 1.5 mm, the magnetic flux passes through the MRF working gap from the left- and right-side guide discs, the flux in the gap varies relatively evenly with the radial distance, and the flux through the MRF is relatively low when it is close to the coil due to the relatively small magnetic resistance of the friction disc compared to the MRF. In Figure 8, we can see that

the local area of the squeeze disc is close to magnetic saturation, and the maximum current is set to 3 A to avoid the heating phenomenon caused by the excessive magnetic flux.

To establish the relation between current, $I$, and MRF shear yield stress before squeeze, $\tau_y(B)$, the energizing current of the coil is varied and the magnetic field strength excited by the current is calculated at a current $I = 0$ to 3.0 A and a gap thickness of 1.5 mm, as shown in Figure 9.

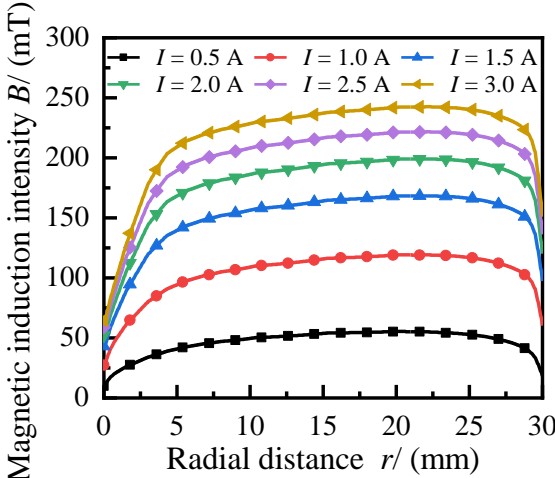

**Figure 9.** Relations among current, working gap radial distance, and magnetic induction intensity.

The radial distance from any plane of the MRF to the small diameter is defined as $r$. As seen from Figure 9, the magnetic induction strength rises rapidly with the increase of current when the current is small, and the growth rate is slow at the current close to 3 A. Since the magnetic induction intensity is small at the two ends of the working gap, the pressurization process of the MRF in the whole gap needs to be considered.

### 2.5. Electrothermal SMA Spring Drive Characteristics

Figure 10a illustrates the physical model of the electrothermal SMA coil spring, and a simplified model based on the sample is shown in Figure 10b. In the simplified model of the electrothermal SMA spiral spring, the diameter of SMA wire is $d = 2.3$ mm, the helix angle of the spring helix is $\alpha = 6°$, the diameter of the spring is $D = 10.6$ mm, the free height of the spring is $L = 21$ mm, and the number of effective spiral turns is $n_e = 6$. The upper and lower ends of the spring are closed and ground, the diameter of the copper-core polyurethane enameled wire spirally wound on the SMA wire is $d' = 0.4$ mm, and the number of turns of the enameled wire is $N_1 = 300$.

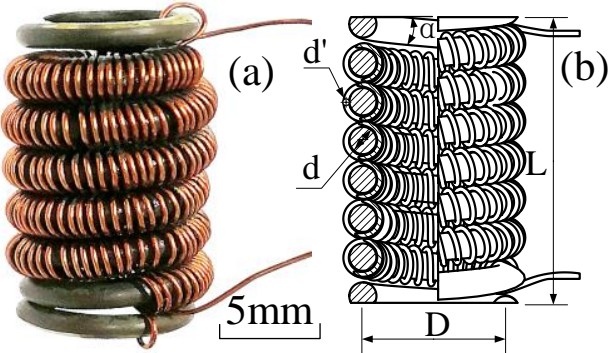

**Figure 10.** Schematic diagram of electrothermal SMA spring. (**a**) Sample diagram and (**b**) principal diagram.

In this transmission device, the main reliance is on the SMA's restoring force to the MRF for squeeze. The specific material parameters of the electrothermal SMA spring are shown in Table 3 [16].

**Table 3.** Material parameters of SMA ($Ni_{51}Ti_{49}$ (at.%)) spring.

| $A_S$/°C | $A_P$/°C | $A_F$/°C | $G_A$/GPa |
|----------|----------|----------|-----------|
| 63.28 | 83.53 | 92.91 | 200 |
| $M_S$/°C | $M_P$/°C | $M_F$/°C | $G_M$/GPa |
| 36.21 | 43.81 | 49.47 | 70 |

$M_S$, $M_F$, $A_S$, and $A_F$ are the starting and finishing transformation temperatures of martensite and austenite, respectively, as shown in Table 3. $G_M$ and $G_A$ are the shear moduli of martensite and austenite, respectively. The SMA 3D fine-scale intrinsic equation under the pure shear mode can be expressed as [22]:

$$\gamma_S = \frac{\tau}{G(K)} + \frac{\tau_0}{G(K_0)} + \frac{\sqrt{6}}{2}\varepsilon_L(\beta - \beta_0) + \gamma_0 \tag{10}$$

where all variables without subscripts are current variables, and those with zero subscripts are the initial values. $K$ is the current temperature of the SMA spring, $\varepsilon_L$ is the maximum residual strain, and $\beta$ is the shape memory factor.

During loading of an electrically heated SMA coil spring, the direction of the force is along the axis of the spring, so that the SMA wire is always subject to a pure shear load. The distance from the center of the SMA wire section to the outer edge of the section is denoted by $x$, and the linear shear strain of the SMA wire section can be expressed as:

$$\gamma_S = \theta x, x \in (0, r_S) \tag{11}$$

where $\theta$ refers to the angle of twist of the SMA spring wire, and $r_S$ refers to the radius of the SMA spring wire.

The deformation of the spring can be expressed as:

$$f = L - L_0 = 2\pi n_e \theta R^2_{SMA} = \frac{\pi n_e \theta D^2}{2} \tag{12}$$

By combining Equations (10) and (12), the axial deformation of the electrothermal SMA coil spring can be obtained as:

$$f = \frac{\pi n_e D^2}{d}\left(\frac{\tau}{G(K)} + \frac{\tau_0}{G(K_0)} + \frac{\sqrt{6}}{2}\varepsilon_L(\beta - \beta_0) + \gamma_0\right) \tag{13}$$

where $\tau$ refers to the shear stress of the SMA wire section, and the shear stress of the coil spring is:

$$\tau = k\frac{8FD}{\pi d^3} \tag{14}$$

where $F$ refers to the axial load of the coil spring, and $k$ refers to the stress correction factor, which can be expressed as:

$$k = \frac{4C_S - 1}{4C_S - 4} + \frac{0.615}{C_S} \tag{15}$$

where $C_S$ refers to the coiling ratio of the coil spring ($C_S = D/d$).

By combining Equations (13) and (10), the axial displacement produced by the electrothermal SMA coil spring under the axial load, *F*, can be obtained as:

$$f = k\frac{8n_eFD^3}{d^4G(K)} - \frac{\pi n_eD^2}{d}\left(\frac{\tau_0}{G(K_0)} + \frac{\sqrt{6}}{2}\varepsilon_L(\beta - \beta_0) + \gamma_0\right) \tag{16}$$

The squeeze force generated by the electric SMA spring under axial displacement is obtained by organizing Equation (16) as:

$$F_r = \frac{d^4G(K)}{8kn_eD^3}f + \frac{\pi d^3G(K)}{8kD}\left(\frac{\tau_0}{G(K_0)} + \frac{\sqrt{6}}{2}\varepsilon_L(\beta - \beta_0) + \gamma_0\right) \tag{17}$$

## 3. Analysis of MR Transmission Device Torque

### 3.1. Pressurization Process of Electrothermal SMA

The relations between the shear modulus, *G*, and temperature of an electrothermal SMA spring can be obtained as [23]:

$$G(\phi_m) = \frac{E}{2(1+v)} = G_A + (G_M - G_A)\phi_m \tag{18}$$

where $\phi_m$ is the martensitic volume fraction, and *E* and *v* are the elastic modulus and Poisson's ratio of SMA, respectively.

The rate of change of the martensitic volume fraction during the inverse martensitic transformation is:

$$\frac{d\phi_m}{dt} = -\frac{\pi\frac{dK}{dt}}{2(A_F - A_S)}\sin\left(\pi\frac{K - A_S}{A_F - A_S}\right) \tag{19}$$

The elongation of the SMA can be approximated to be zero due to the small deformation in the MRF squeeze process. Bringing Equation (18) into Equation (17), the relationship between the spring squeeze force and the current is shown in Figure 11.

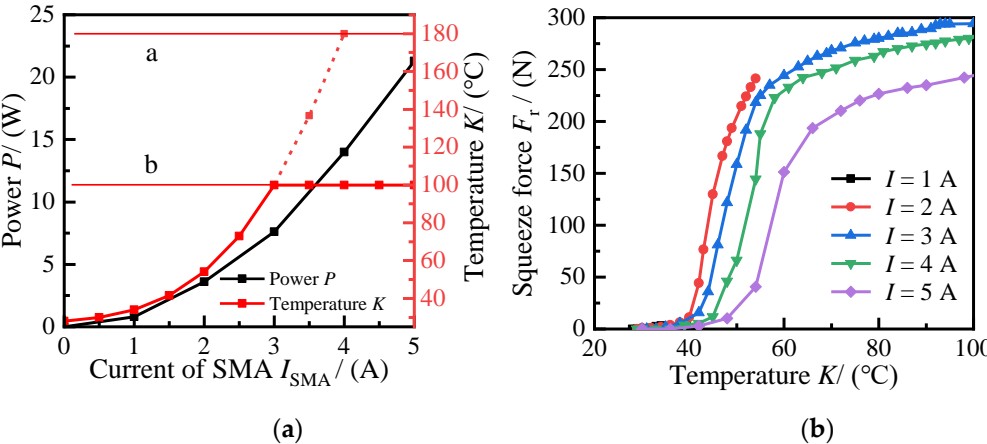

**Figure 11.** Electrothermal characteristics curve of SMA. (**a**) Relationship among current, power, and spring temperature. (**b**) Relationship between squeeze force and temperature.

The solid line in the current versus temperature curve in Figure 11a represents the actual recorded test data, while the dashed line shows theoretical current versus temperature values. The temperature corresponding to line b in the figure is 100 °C, indicating the maximum response force that SMA can generate, and the temperature corresponding to line a in the figure is 180 °C, indicating the ultimate operating temperature of the copper-core enameled wire. The experimental values in Figure 11b illustrate the response force corresponding to the electrothermal SMA at various currents and steady-state temperatures,

with the graph showing that the maximum response force of the spring increases as the current increases.

As seen from Figures 11b and 12, when the current is greater than 3 A, the martensitic phase transformation is accelerated due to the excessive heat generated by the enameled wire, resulting in an unstable metallurgical structure and a tendency for the output force to decrease [24]. While the current increases, the spring's recovery rate decreases and the temperature interval for deformation recovery shifts back.

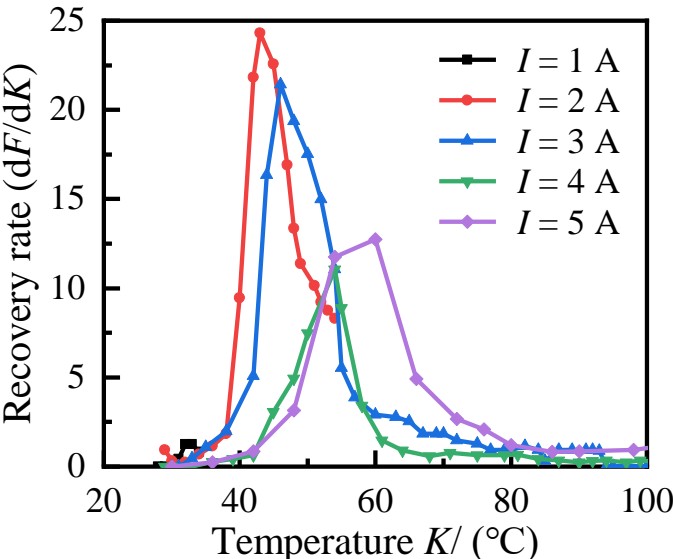

**Figure 12.** Relationship between recovery rate and temperature.

As the SMA temperature rises from 40 to 65 °C during the heating-induced shape memory effect stage, the martensite content decreases and the spring returns to its initial state. As the axial displacement of the SMA spring is limited, the strain energy is converted to mechanical energy and used to generate the axial force.

### 3.2. Quantitative Calculation of Transmission Performance

It is assumed that the all the MRF in the working gap yields and performs shear flow under the action of the magnetic field, and the torque calculation formula of the disc MR drive is as follows [7]:

$$
\begin{aligned}
M_\mathrm{M} &= \int_{r_2}^{r_1} 2\pi\tau r^2 dr \\
&= \frac{2\pi}{3}\left(r_1^3 - r_2^3\right)\tau_y(B) + \frac{\pi}{2h}\left(r_1^4 - r_2^4\right)(\omega_1 - \omega_2)\eta
\end{aligned}
\tag{20}
$$

where $\eta$ refers to the zero-field viscosity, $h$ refers to the thickness of the MRF gap, $r_1$ and $r_2$, respectively, refer to the large and small diameter of the working gap, and $\omega_1$ and $\omega_2$ refer to the angular velocity of the input and output shaft, respectively. $\omega_1$ and $\omega_2$ are 100 and 80 rad/s, respectively, which means the shear strain, $\gamma$, is 0.2.

According to the squeeze experiment of MRF [19], in the case of $B_0$ = 50 mT, $\Delta$ = 20 mT, and magnetic particles spaced far apart ($R/d_0$ < 0.5), we set the squeeze deformation constants as X = 0.01 and Y = 0.4. The deformation relationship of magnetorheological fluid according to Equations (6) and (8) is shown in Figure 13.

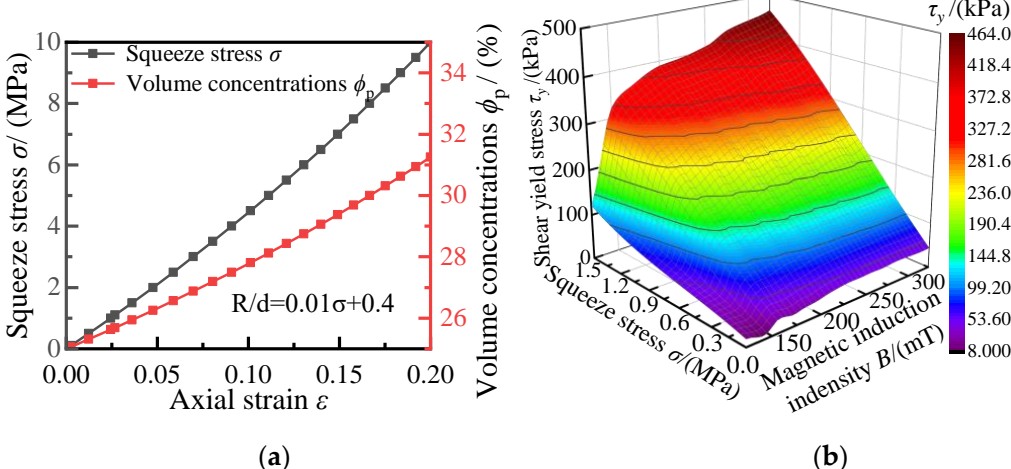

**Figure 13.** Squeeze characteristics of the MRF-J01T. (**a**) Deformation relationship of the MRF. (**b**) Relationship between squeeze stress and shear yield stress.

　　Bringing the data of Figure 13 into Equation (20), the composite drive torque at the different currents of the SMA can be calculated, as shown below. Figure 14 depicts the effect of the electrothermal SMA current on the torque of the transmission for a coil current of 3 A. When the spring current exceeds 1 A, the torque of the transmission rapidly increases in an approximately linear manner, from 15.08 to 73.56 N·m, a 4.88-fold increase. Eight SMA springs provided a total squeeze stress of 0.34 MPa.

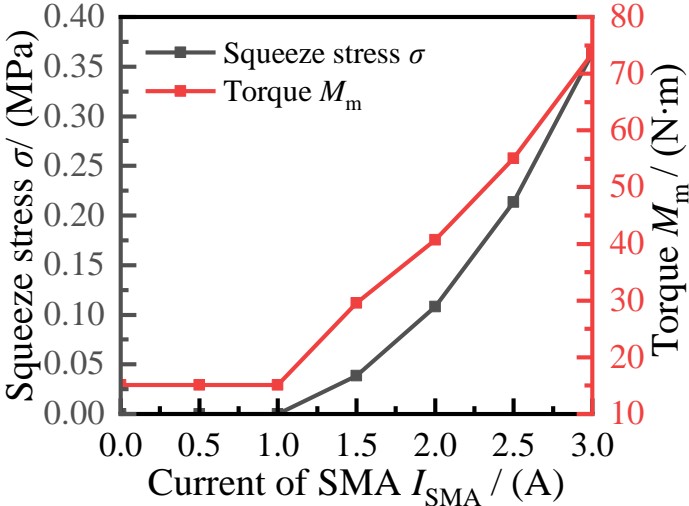

**Figure 14.** Torque of the transmission device before and after squeeze.

## 4. Discussion

　　The SMA springs could be thermally elongated by using enameled wire energized with heat, which could effectively output the squeezing force. Active control of the SMA spring has been achieved. From the experiment, we found that the steady-state temperature of SMA grows rapidly with the increasing power supply, while the output force gradually increases. However, when the current is higher, the martensite phase transformation is too fast, which easily leads to the problem of unstable metallurgical transformation and weakened output force, so how to reasonably control the phase transformation temperature of the SMA will be the focus of further research.

　　Furthermore, the shear yield stress of MRF is primarily caused by two parts: the dipole moment between magnetic dipoles and the friction of magnetic particles. According to the squeeze model established in this article, the shear yield stress was mainly related to

the squeeze stress and shear strain, and the squeeze strengthening effect of MRF would disappear when the shear strain was too large. Nonetheless, as the squeeze stress increases, the percentage of shear stress due to particle friction increases, and the shear yield stress is further enhanced (in a small deformation range).

By using electrothermal SMA spring pressurization, the performance of the transmission has risen by 4.88 times, solving the current problem that the torque of the MR transmission device is small. Moreover, when compared to the existing squeezed MR device, the composite device has a simpler structure and a high energy utilization, providing a new design idea for the squeezed MR transmission device.

As the squeeze model used in this article was based on steady-state pressurization, it failed to consider the effect of transient shocks on the transmission torque. The actual process of transforming the single-chain model to the BCT model was more complicated, which limited the accuracy of the torque equation to a certain extent. In the future, the variation from a single-chain to a BCT structure on any short timescale could be simulated in detail, and the theoretical accuracy would be improved.

## 5. Conclusions

This paper proposed a variable thickness MR transmission device controlled by electrothermal SMA springs, which realizes a double-disc transmission and employs an electrothermal SMA spring to change the MRF working gap thickness and control the transmission torque. This paper also performed a theoretical analysis of the magnetic circuit and magnetic field distribution of the MRF transmission device, and a theoretical deduction and calculation of the squeezing force output by the electrothermal SMA spring and the transfer torque of the variable thickness MRF transmission device. The findings indicated that:

(1) The output squeeze force of the SMA spring showed a high degree of nonlinearity during the temperature rise, with a maximum squeeze force of 318.43 N. When the current of the electrothermal SMA spring was 3 A, the highest heating temperature of the spring could reach the end of austenite phase transformation temperature. The current size affects the phase transformation process of the SMA to some extent, resulting in differences in the growth trajectory of the spring's restoring force.

(2) Thickness variation had a small effect on the integral number of MRF, but excessive shear deformation leading to magnetic chain breakage was the main factor affecting the performance failure of MRF. The shear stress under squeeze conditions was mainly composed of two parts: magnetic shear stress of dipole and friction stress between particles. When the MRF was axially pressurized, the magnetic chain gradually changed from a single-chain model to a body-centered cubic model, along with the reduction of the particle gap. The squeeze strengthening effect of MRF is related to the magnetic induction strength and squeezing pressure. When the squeezing pressure is fixed, the higher the magnetic induction strength, and the more significant the strengthening effect.

(3) The torque of MR transmission device showed an approximately linear increase with the increase of the spring current, and the maximum MRF torque was 73.56 N·m at the current of 3.0 A and the squeeze stress of 0.34 MPa, and the performance of MRF was enhanced by 4.88 times after squeeze strengthening.

**Author Contributions:** J.H. and S.C. conceived this research; W.C. and S.C. performed the calculations; W.C. and S.C. wrote the original draft of the manuscript; S.C. and J.H. edited and reviewed the manuscript; J.H. oversaw the progress of the study and the visualization of the results. All authors have read and agreed to the published version of the manuscript.

**Funding:** This research was funded by the National Natural Science Foundation of China under Grant No. 51875068, the Youth Project of Science and Technology Research Program of Chongqing Education Commission of China under Grant No. KJQN201901120, and the Natural Science Foundation Project of Chongqing Science and Technology Commission (cstc2020jcyj-msxmX0402).

**Institutional Review Board Statement:** Not applicable.

**Informed Consent Statement:** Not applicable.

**Data Availability Statement:** Not applicable.

**Acknowledgments:** The authors gratefully acknowledge the financial support of this work from the National Natural Science Foundation of China under Grant No. 51875068, the Youth Project of Science and Technology Research Program of Chongqing Education Commission of China under Grant No. KJQN201901120, and the Natural Science Foundation Project of Chongqing Science and Technology Commission (cstc2020jcyj-msxmX0402). Finally, the authors would like to thank the editors and reviewers for their valuable comments and constructive suggestions.

**Conflicts of Interest:** The authors declare no conflict of interest. The funders had no role in the design of the study; in the collection, analyses, or interpretation of data; in the writing of the manuscript, or in the decision to publish the results.

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
