# Peer review of "Study of Variable Thickness Magnetorheological Transmission Performance of Electrothermal Shape Memory Alloy Squeeze"

_applsci, doi:10.3390/app12094297_

Round 1

Reviewer 1 Report

In this work, the authors studied a variable thickness MRF transmission device controlled by an electrothermal SMA spring. The authors made a theoretical analysis of the magnetic circuit and magnetic field distribution of the MRF transmission device, and further made a theoretical deduction and calculation of the squeezing force output by the electrothermal SMA spring and the transfer torque of the variable thickness MRF transmission device. The results were analyzed systematically and discussed in depth. Overall, this work is of good scientific soundness and worthy to be published after minor revision.

Comments to Author:

  1. Please improve the figure quality of figure 1 by using higher resolution
  2. Please improve the figure quality of figure 12 by making the text in legend larger
  3. In the conclusion part, could the authors compare the performance of their proposed device to other reported state-of-art literatures?

Reviewer 2 Report

The paper addresses a theoretical analysis of a MRF coupling providing a high torque density by using the squeeze strengthening effect. A shape memory alloy (SMA) spring is used for compression, combining various smart materials to create an advantageous design for the MRF coupling.

Unfortunately, the purely theoretical approach does not do justice to the interesting topic of the article. The highly simplified approach to modelling the squeeze-boost effect leads to a torque prediction that cannot be validated in an experimental verification. The significant reduction of the strengthening effect under the influence of shear was not mentioned or considered in any way, as reported e.g. in the cited literature [12]. Moreover, the parallel connection of the excitation coil with the wire of the SMA spring does not appear to make sense and contradicts the functional principle described. Likewise, the magnetic design of the clutch has significant lacks that require a fundamental revision.

The Conclusions in section 6 of the investigation describe the well-known behaviour of SMA springs as well as the linear relationship between magnetic excitation and magnetic flux density in case of unsaturated magnetic circuit, and add, thus, no gained knowledge.

Due to the significant deficiencies and the errors listed in detail below, a publication of this paper cannot be recommended.

Further criticism in detail

The mechanical design presented in Figure 1 and Figure 2 has various inconsistencies. Although a sealing ring is shown in the inner area of Figure 1, the design shown requires another sealing ring in the outer area. The design depicted in Figure 1 and Figure 2 only allows the conclusion that a relative movement between input and output shaft is impossible. The identification of the MRF in Figure 1 is incorrect.

The assignment of the magnetic permeability in Equation 1 is completely wrong. The same magnetic permeability is assigned to the housing and the shear gap filled with MRF and the vacuum permeability is not included. Figure 2 uses uppercase letters, Equation 1 includes lowercase letters, a reference is not made.

The assignment of materials on page 5 is not complete. What material is used for the rotary housing? For what reason different steels are used for input shaft and extrusion disk?

The unit of magnetic resistance on page 5 is missing. Likewise, the units in Equation 6 are incorrect.

The insufficient resolution of Figure 5 does not allow any meaningful interpretation. The caption and the scale of Figure 5 are wrong. It is probably the representation of the magnetic flux density and the flux lines. Scale A in Weber has no context related to Figure 5. Due to the different scales of the magnetic flux density, the comparability of the representation suffers. The high maximum values of the magnetic flux density of more than 4 Tesla suggest an incorrect meshing. On page 6, conclusions are drawn from Figure 5 about shear gap heights which are not shown in the referenced figure.

The resolution of Figure 6 is insufficient and the caption is incorrect, as mentioned before for Figure 5.

The parameter R shown in Figure 7 is not introduced and does not allow an interpretation of the Figure.

The source [18] used to introduce Equation 8 deals with magnetorheological elastomers and not SMA. The variables used in Equation 8 are not introduced.

Equation 16 appears to result from Equation 15 and not from Equation 12.

Equation 17 uses inconsistent units.

The section on the squeeze strengthening effect is not detailed enough and does not address the effects mentioned in the literature listed. In particular, the breakdown/reduction of the squeeze strengthening effect under the influence of shear is not addressed and considered.

In Equation 19 the reference to the superimposed torque is missing, also the units are inconsistent.

Reviewer 3 Report

The paper describes a composite transmission device by extruding magnethorheological fluid with electrothermal shape memory alloys spring. The devise is using for improving the transmission of torque.

In the abstract, please, replace “Research results:” with a full sentence. Also add between brackets MRF and SMA when it was described for the first time in the abstract.

The state of the art could be extended with at least 5 references with less than 5 years (from 2017 up to now). As an example, your recent publication in Applied Sciences (Research on the Transmission Performance of a High-Temperature Magnetorheological Fluid and Shape Memory Alloy Composite)

More Introduction section comments:

  • field[2-3]. Shape memory ally (SMA). There is a typo, change ally by alloy.
  • type clutches, The transmittable torque. Please, add a point instead comma.
  • The electro-thermo-mechanical haracterization of SMA. Complete the word “haracterization” as “characterization”
  • Revise this sentence: “Hwang et al[15] to achieve”
  • Last paragraph: rewrite in more suitable form. Avoid to use “the team of the Authors” and describe it in an affirmative way, not as a challenge.

The paper is not well structured. I recommend authors to reorder and rewrite the paper. In order to be easier to follow for readers I recommend to authors to add a section called 2. Materials & Methods starting with a brief description of the different used methods in a flowchart or image. In this way, authors should add the actual sections 2 to 5 inside them (ie, 2.1 Design of variable thickness MRF transmission device / 2.1.1 Working principles and so on).

Results, both FEM and theoretical models, are mixed with other sections. I recommend to authors to separate them and put them in this section, with different sub-sections.

It is needed a Discussion section prior to Conclusions section. Author have not discussed the results and findings.

Although conclusions are supported by results, I recommend to authors to add the limitations of the study, but also the different applications of the research. Also, I recommend to add more future works derived from this research.

Figures 1 to 4 and 9, increase the quality of the image.

Figure 5 and 6, increase the quality. I suggest to authors to use vertical table for the three images and increase the size. The text (numbers mainly) should be readable with higher quality than now.

Round 2

Reviewer 3 Report

Authors have adressed very well some comments, but there are some of them not well accomplished:

  • The state of the art could be extended with at least 5 references with less than 5 years (from 2017 up to now). As an example, your recent publication in Applied Sciences (Research on the Transmission Performance of a High-Temperature Magnetorheological Fluid and Shape Memory Alloy Composite)

  • Authors have added Materials and Method section. But the section "2. Working principles of variable thickness MRF transmission device", should be inside this section, as 2.1. Take into account also the next point.
  • Results, both FEM and theoretical models, are mixed with other sections. I recommend to authors to separate them and put them in this section, with different sub-sections.

  • It is needed a Discussion section prior to Conclusions section. Author have not discussed the results and findings. There is not discussion in relation with the state of the art. If authors include in conclusions section the Discussion, I recomend to change this section name to Discussion and Conclusions.

Round 3

Reviewer 3 Report

The authors have complied with all comments, although the results are mixed with the methods, I think the research is well understood. With the discussion section the manuscript has been significantly improved.